High microvessel and lymphatic vessel density predict poor prognosis in patients with esophageal squamous cell carcinoma

Qi Li Wen 1 2
Xie Yu Fang 1 3
Wang Wei Nan 1
Liu Jia 1
Yang Kai Ge 1
Chen Kai 1
Luo Cheng Hua 1
Fei Jing 2
Hu Jian Ming 1 jianming.120@163.com
1 Pathology, Shihezi University School of Medicine/The First Affiliated Hospital of Shihezi University , Shihezi, Xinjiang , China
2 Department of Oncology, The First Affiliated Hospital of Shihezi University , Shihezi, Xinjiang , China
3 Department of Pathology, The Affiliated Cancer Hospital of Zhengzhou University & Henan Cancer Hospital , Zhengzhou, Henan , China
Camacho-Villegas Tanya
Electronic publication date: 2024 Sep 27
Publication date: 2024
Volume: 12
Electronic Location ID: e18080
Received 2024 May 3; Accepted 2024 Aug 20
Copyright: © 2024 Qi et al.
Copyright year: 2024
Copyright holder: Qi et al.
License: This is an open access article distributed under the terms of the Creative Commons Attribution License, which permits unrestricted use, distribution, reproduction and adaptation in any medium and for any purpose provided that it is properly attributed. For attribution, the original author(s), title, publication source (PeerJ) and either DOI or URL of the article must be cited.
License URL: https://creativecommons.org/licenses/by/4.0/

Keywords: Esophageal squamous cell carcinoma, Microvessel density, Lymphatic vessel density, Prognosis

Funding: Xinjiang Production and Construction Corps 2023AB058, 2022ZD003, 2023ZD028, and 2023ZD027 Shihezi University GJHZ202001, LJ202202, ZZZC2023029, and MSPY202407 Rural Area in China 2024 This study was funded by Key scientific and technological research projects in the Xinjiang Production and Construction Corps (No. 2023AB058), Guiding scientific and technological research project in the Xinjiang Production and Construction Corps (No. 2022ZD003, No. 2023ZD028, No. 2023ZD027), International Science and technology cooperation promotion project of Shihezi University (GJHZ202001), Clinical Basic Research Project of the First Affiliated Hospital of Shihezi University (No. LJ202202), National Early Detection and Treatment Project for Upper Digestive Tract in Rural Area in China (NO. 2024), Scientific research project of early gastrointestinal cancer doctors’ joint growth plan (GTCZ-2023-XJ-02), Shihezi University independently funded research and innovation projects (ZZZC2023029). (NO. 2024), Scientific research project of early gastrointestinal cancer doctors’ joint growth plan (GTCZ-2023-XJ-02), Shihezi University independently funded research and innovation projects (ZZZC2023029). The funders had no role in study design, data collection and analysis, decision to publish, or preparation of the manuscript.

==============================
Background

Microangiogenesis and lymphangiogenesis are essential for tumor growth in the tumor microenvironment, contributing to tumor invasion and metastasis. Limited literature exists on these processes in esophageal squamous cell carcinoma (ESCC). Therefore, the purpose of this study is to explore the impacts of microangiogenesis and lymphangiogenesis on the occurrence, progression, and prognosis assessment of ESCC.

Methods

Surgical specimens and paraffin-embedded human tissues were procured from ESCC patients, encompassing 100 ESCC tissues and 100 cancer-adjacent normal (CAN) tissues. CD34 and D2-40 were utilized as markers for microvessel endothelial cells and lymphatic vessel endothelial cells, respectively. Microvascular density (MVD) and lymphatic vessel density (LVD) were evaluated through immunohistochemical quantification.

Results

We found that tumor tissues in ESCC patients had significantly higher MVD and LVD than cancer-adjacent normal (CAN) tissues. High MVD and LVD were associated with lymph node metastasis and advanced tumor clinical stages. Additionally, both high MVD and high LVD were strongly linked to poorer prognosis among cancer patients. Furthermore, a positive correlation was found between high MVD and high LVD (p < 0.05). The presence of these markers individually indicated a worse prognosis, with their combined assessment showcasing enhanced prognostic value.

Conclusions

Overall, the increased MVD and LVD indicates higher invasion and metastasis of ESCC, closely correlating with unfavorablefor poor prognosis of ESCC patients.

Introduction

Esophageal carcinoma (ESCA) is a prevalent malignant gastrointestinal tumor, characterized by its heterogeneous disease with high morbidity and mortality that occurs in the epithelial cells of the esophagus. It is the eighth most common malignant tumor in the world and the sixth leading cause of cancer-related mortality (Siegel, Giaquinto & Jemal, 2024). ESCA encompasses two main types: adenocarcinoma and esophageal squamous cell carcinoma (ESCC), with the latter comprising the majority of cases worldwide (Zhang, Wang & Meng, 2022). The early symptoms of ESCA are obscure, once detected, the disease would have developed to the intermediate and late stages (Fatehi Hassanabad et al., 2020). Advanced ESCA typically presents with local infiltration and lymphatic metastasis, contributing significantly to its poor prognosis (Jones, 2020). Angiogenesis and lymphangiogenesis are the key processes of tumor invasion and metastasis (Liu et al., 2023; Rezzola et al., 2022). Tumor neovascularization exhibits an incomplete structure, thin vessel walls, and absence of basement membrane in most cases, facilitating tumor cell entry into blood circulation for metastasis through neovascularization (Qian & Pollard, 2010). Microvessels (MV) serve as markers of angiogenesis in tumor tissue and form the basis for tumor cell migration and invasion. Microvessels density (MVD) has become a reference index for evaluating invasive ability and prognostic impact across various tumors (Giuppi et al., 2021; Guo et al., 2015). CD34 is a specific marker of endothelial cells used to detect MVD as well as a mature marker of tumor neovascularization within this context (Grizzi et al., 2023). While angiogenesis plays a crucial role in tumor growth and progression, many scholars emphasize that lymphatic metastasis is equally important in determining survival outcomes while also influencing treatment strategies selection for malignant tumors. Several novel, specific antibodies targeting lymphatic endothelial cells have been developed and utilized in the investigation of human cancer lymphangiogenesis (Dieterich et al., 2022). The antibody D2-40 is employed to detect the fixed resistance epitope on podoplanin, a selective marker of lymphoendothelium that can be utilized to identify LVs in formalin-fixed, paraffin-embedded tissue sections. The assessment of lymphatic vessel density (LVD) using D2-40 is another crucial evaluation index for cancer invasion, metastasis, and prognosis.

Invasion and metastasis play crucial roles in the recurrence and mortality of patients with malignant tumors. Microangiogenesis and lymphangiogenesis within the tumor microenvironment are essential for promoting tumor growth and metastasis. However, there is limited research on the relationship between microangiogenesis, lymphangiogenesis, and tumor progression in ESCCs. This study aims to assess the significance of microvascular and lymphatic markers in the occurrence, progression, and prognosis of ESCCs.

Materials and Methods

Patients and specimens

Surgical specimens and paraffin-embedded tissues were derived from ESCC patients, and included 100 ESCC and 100 cancer-adjacent normal (CAN) tissues. None of the preoperative tumor patients had undergone radiotherapy or chemotherapy. Between 2018 and 2021, we conducted research on ESCC patients at the Department of Pathology, First Affiliated Hospital of Shihezi University in Xinjiang, China. Based on the World Health Organization histological tumor classification criteria, two pathologists confirmed a diagnosis of ESCC. According to the National Comprehensive Cancer Network (NCCN) guidelines (Ajani et al., 2023), 67 patients were clinically staged I–II and 33 had stage III–IV disease. The survival status of all patients was ascertained via telephone until December 2022. The median follow-up for surviving patients was 30 months (range, 1–78 months). Overall survival (OS) was defined as the interval between surgery and death or between surgery and the last follow-up in surviving patients. Each participant provided written, informed consent before registering for the study. The study was approved by the Science and Technology Ethical Committee of the First Affiliated Hospital, Shihezi University School of Medicine (201803801) in accordance with the ethical guidelines of the Declaration of Helsinki.

Immunohistochemistry

Paraffin sections were cut (4-μm thick) and deparaffinized with xylene. The sections were dehydrated in xylene and rehydrated using a graded series of ethanol solutions. After dewaxing, the sections were incubated in 3% peroxide-methanol solution at room temperature, and then repaired at 100 °C. The sections were incubated at room temperature for 30 min, and then washed with phosphate-buffered saline (PBS). Then, the sections were incubated with antibodies against CD34 (mouse monoclonal, 1:400; Zsbio) and D2-40 (mouse monoclonal, 1:100; Zsbio) for 8 h. Subsequently, the sections were washed thoroughly with PBS, and primary antibody binding was visualized using an EnVision kit (DAKO, Glostrup, Denmark) following the manufacturer’s instructions. Finally, the sections were faintly counterstained with hematoxylin and mounted with glycerol gelatin.

Immunoreactivity evaluation

CD34 was used as a microvascular endothelial cell marker for the MVD and MV count; CD34 positive expression in the cell membrane and/or cytoplasm appeared as brownish yellow particles. MVs were identified as follows: CD34-stained endothelial cells were surrounded by blood vessels, with a diameter of less than eight red blood cells are counted as MVs. The immunostained sections were scanned under light microscopy (OLYMPUS BX51) at low magnification (40×) and the areas with the greatest number of distinctly highlighted MVs (“hot spots”) were selected. MVD was determined by counting all immunostained MVs at 200× magnification. Five hot spots were chosen in each section. The mean values of the MVs in the five fields were analyzed statistically.

D2-40 was used as a lymphatic endothelial cell marker for the LVD; D2-40 positive expression was observed in the cytoplasm as brownish yellow particles. The immunostained sections were scanned under light microscopy (OLYMPUS BX51) at low magnification (40×), and hot spots with the greatest number of distinctly highlighted LVs were selected. LVD was determined by counting all immunostained LVs at 200× magnification. Five hot spots were chosen in each section (Sharma et al., 2023). The mean values of the LVs in the five fields were analyzed statistically.

Statistical analysis

All statistical analyses were performed using SPSS version 20 (IBM, Endicott, NY, USA). The difference in CD34 and D2-40 expression between the ESCC and CAN groups was compared using a nonparametric test. The clinical parameters between the high MVD and LVD group and low MVD and LVD group were compared using the χ2 test. For classification analysis, mild expressions were divided in to two groups according to CD34 and D2-40 expression. The correlation between MVD and LVD co-expression, and the ESCC clinical parameters, were evaluated using Spearman rank correlation. Survival curves were estimated using the Kaplan–Meier method. The expression of both CD34 and D2-40, the correlation between clinicopathological parameters, and the OS rate were evaluated in univariate analysis. Multivariate analysis was performed using a Cox proportional risk regression model. The p-value was calculated using EPi Info; p < 0.05 was considered significant.

Results

ESCC tissue exhibits greater MVD and LVD than CAN tissues

CD34 and D2-40 were used as MV and LV endothelial cell markers, respectively (Figs. 1, 2). Immunohistochemistry staining was used to demonstrate the distribution of MVs and LV in ESCC, and the MVD and LVD were quantified and analyzed. The average MVD in ESCC tissues was 16/HPF (high-power fields) (range, 3–33/HPF), which was significantly higher than that in CAN tissues (range, 2–19/HPF) (p < 0.001). The average LVD in ESCC tissues was 6/HPF (range, 2–14/HPF), which was also significantly higher than that in CAN tissues (range, 0–7/HPF) (p < 0.001) (Fig. 3).

Figure 1 Immunohistochemical staining of MVD in patients with ESCC.

CD34 positive expression is observed in the cell membrane and/or cytoplasm as brown-yellow particles. The CD34 positive marker clearly shows the distribution characteristics of the MVs. (A) Low-MVD area (100× magnification). (B) Low-MVD area (200× magnification). (C) High-MVD area (100× magnification). (D) High-MVD area (200× magnification).

Figure 2 Immunohistochemical staining of lymphatic vessel density (LVD) in patients with ESCC.

D2-40 positive expression is observed in the cell membrane and/or cytoplasm as brown-yellow particles. The D2-40 positive marker clearly demonstrates the distribution characteristics of the lymphatic vessels (→). (A) Low-LVD area (100× magnification). (B) Low-LVD area (200× magnification). (C) High-LVD area (100× magnification). (D) High-LVD area (200× magnification).

Figure 3 The distribution of microvessels and lymphatic vessels in ESCC tissues is significantly higher compared to that in the normal esophageal tissues.

(A) ESCC tissues had significantly higher MVD than the CAN tissues (****p < 0.001). (B) ESCC tissues had significantly higher LVD than the CAN tissues (****p < 0.001).

Correlation between the distribution of MV and LV and the clinicopathological characteristics of ESCC

To analyze the role of MV and LV in ESCC progressions, the clinicopathological parameters of the patients were collected and tissue sections were examined. Patients with lymphatic metastasis had significantly higher MVD compared with patients without lymphatic metastasis (p < 0.05). Patients with advanced ESCC had significantly higher MVD than patients with early-stage ESCC (p < 0.05). At the same time, patients with lymphatic metastasis had significantly higher LVD than patients without lymphatic metastasis (p < 0.05). Moreover, LVD in advanced ESCC was significantly higher than that in early-stage ESCC (p < 0.05). There was no significant correlation between the distribution of MV, LV and age, sex, differentiation, invasion depth (all, p > 0.05) (Table 1).

Table 1 Correlation between microvessel density (MVD), lymphatic vessel density (LVD) and clinicopathological features of esophageal squamous cell carcinoma.

Variable	Cases (N)	MVD
Mean ± sd.	P	LVD
Mean ± sd.	P	
Age (y)						
≤Median (58y)	53	16.02 ± 8.02	0.623	5.75 ± 2.33	0.165	
>Median	47	16.74 ± 6.50		6.53 ± 3.20		
Gender						
Male	66	17.23 ± 7.46	0.099	6.48 ± 2.97	0.068	
Female	34	14.68 ± 6.81		5.41 ± 2.27		
Histologic grade						
Well	32	16.72 ± 8.28	0.180	6.16 ± 2.62	0.893	
Moderate	48	15.15 ± 6.39		6.00 ± 2.71		
Poor	20	18.70 ± 7.55		6.35 ± 3.33		
Depth of invasion						
T1-T2	40	15.68 ± 7.11	0.447	6.45 ± 3.08	0.336	
T3-T4	60	16.82 ± 7.48		5.90 ± 2.58		
lymphatic metastasis						
Negative	52	14.65 ± 7.40	0.014**	5.48 ± 2.35	0.016*	
Positive	48	18.21 ± 6.83		6.81 ± 3.07		
TNM stage						
I–II	67	14.93 ± 6.98	0.005**	5.64 ± 2.62	0.014*	
III–IV	33	19.27 ± 7.23		7.09 ± 2.89		
Notes:

TNM, Tumor Node Metastasis.

* P < 0.05

** P < 0.01.

Correlation between MVD and LVD and their synergistic role in the evaluation of clinical progression of ESCCs

To analyze the association between MVs and LVs in patients with ESCC, we performed a correlation analysis of ESCC samples, and found a significant positive correlation between MVD and LVD (p < 0.001) (Fig. 4). The above findings indicate a strong correlation between increases in MVD and LVD with lymph node metastasis and clinical staging of ESCC. There exists a significant linear relationship between MVs and LVs. Is the combined effect of MVD and LVD more valuable for assessing ESCC invasion and clinical progression? Based on the optimal cut-off numbers (MVD: 16, LVD: 6), we divided the cases into four groups: high MVD, high LVD, low MVD and low LVD to evaluate the synergistic effect of MVD and LVD in the invasion and clinical progression of ESCC. Analysis showed that lymph node metastasis and clinical progression were significantly increased in patients with high MVD and low LVD compared with patients with low MVD and LVD (p < 0.05), but were independent of other clinicopathological parameters (sex, age, degree of differentiation, depth of invasion, p > 0.05) (Table 2). It is suggested that the combination of high MVD and high LVD is associated with tumor invasion and lymphatic metastasis, and can be used as an important parameter to evaluate the biological behavior and metastasis trend of ESCC patients.

Figure 4 Correlation between MVD and LVD.

There was a significant positive correlation between MVD and LVD in ESCC patients (r = 0.585, p < 0.001).

Table 2 Correlation between microvascular, lymphatic vessels and clinicopathological features of esophageal squamous cell carcinoma.

Variable	Cases (N)	High MVD and LVD	Low MVD and LVD	X 2	P	
Age (y)						
≤Median (58y)	37	12 (32.4%)	25 (67.6%)	0.035	0.851	
>Median	32	13 (40.6%)	19 (59.4%)			
Gender						
Male	43	18 (41.9%)	25 (58.1%)	1.760	0.185	
Female	26	6 (23.1%)	20 (76.9%)			
Histologic grade						
Well	24	10 (41.7%)	14 (58.3%)	0.608	0.448	
Moderate	35	10 (28.6%)	25 (71.4%)			
Poor	10	4 (40.0%)	6 (60.0%)			
Depth of invasion						
T1-T2	27	9 (33.3%)	18 (66.7%)	0.000	1.000	
T3-T4	42	15 (35.7%)	27 (64.3%)			
Lymphatic metastasis						
Negative	36	7 (19.4%)	29 (80.6%)	6.457	0.011*	
Positive	33	17 (51.5%)	16 (48.5%)			
TNM stage						
I-II	46	11 (23.9%)	35 (76.1%)	5.822	0.016*	
III-IV	23	13 (56.5%)	10 (43.5%)			
Notes:

MVD, Microvessel Density; LVD, Lymphatic Vessel Density; TNM, Tumor Node Metastasis.

* p < 0.05.

Correlation between MVD, LVD and the prognosis of patients with ESCC

To further investigate the prognostic significance of MVD and LVD in patients with ESCC, Kaplan-Meier survival analysis was employed to assess the correlation between MVD, LVD, and overall survival. The findings revealed a significantly lower overall survival period in patients with high MVD compared to those with low MVD (p < 0.05). Similarly, patients with high LVD exhibited significantly lower overall survival period than those with low LVD (p < 0.05).

Moreover, patients with high-density MVD and LVD demonstrated markedly reduced overall survival period compared to those with low-density MVD and LVD (p < 0.001). Overall survival period in non-synergistic groups for MVD and LVD (High MVD + Low LVD or High LVD + Low MVD) fell between that of high-density and low-density groups. However, no statistically significant difference was observed among the three groups (p > 0.05) (Fig. 5). Cox univariate regression analysis indicated that depth of invasion, lymph node metastasis, clinical stage, high MVD, and high LVD were all significantly negatively correlated with the prognosis of ESCC (p < 0.05). Multivariate analysis revealed that only clinical stage emerged as an independent prognostic factor for ESCC patients and displayed a significant negative correlation with overall survival period (p < 0.05) (Table 3).

Figure 5 The correlation between MVD and LVD and clinical prognosis in patients with ESCC.

(A) The high-MVD group had worse prognosis than the low-MVD group (p < 0.05). (B) The high-LVD group had worse prognosis than the low-LVD group (p < 0.05). (C) The high-MVD and high-LVD group had poor prognosis compared with the low-MVD and low-LVD group (p < 0.001). (D) The prognosis of patients with no synergistic of MVD and LVD group (N group) was between patients with high-density MVD, LVD (H group) and patients with low-density MVD, LVD group (L group), but no statistically significant difference between the non synergistic of MVD and LVD group with the other two groups (p > 0.05). Abbreviations: MV, microvessel; LV, lymphatic vessel. Non synergistic group includes high MVD + low LVD, high LVD + low MVD.

Table 3 COX univariate regression analysis of clinical parameters and indicators and prognosis of esophageal squamous cell carcinoma.

Variable	Cases (N)	Univariate analysis	Multivariate analysis	
HR	95% CI	P	HR	95% CI	P	
Age (>58 y/≤58 y)	100	0.862	[0.562–1.321]	0.495				
Gender (female/male)	100	1.043	[0.660–1.651]	0.868				
Histologic grade(Poor/moderate+well)	100	0.848	[0.623–1.152]	0.292				
Depth of invasion (T3 + T4/T1 + T2)	100	1.580	[1.022–2.443]	0.04*	1.253	[0.757–2.074]	0.380	
Nodal metastasis (Positive/negative)	100	2.138	[1.364–3.351]	0.001**	1.307	[0.707–2.418]	0.394	
TNM stage (III + IV/I + II)	100	3.676	[2.197–6.150]	<0.001***	2.505	[1.157–5.420]	0.020*	
Microvessel density	100	1.032	[1.001–1.063]	0.042*	1.104	[0.982–1.048]	0.398	
Lymphatic vessel density	100	1.096	[1.014–1.184]	0.021*	1.026	[0.936–1.124]	0.582	
Notes:

TNM, Tumor Node Metastasis; HR, Hazard Ratio; CI, Confidence Interval.

* p < 0.05.

** p < 0.01.

*** p < 0.001.

Discussion

Angiogenesis and lymphangiogenesis in the tumor microenvironment are crucial for tumor growth. Recent studies have demonstrated that angiogenesis and lymphangiogenesis play a role in tumor progression, making inhibiting these processes within the tumor microenvironment an emerging strategy in anti-cancer therapy (Li et al., 2018; Shojaei, 2012). Microvessel density (MVD) with endothelial markers is considered one of the most useful prognostic markers for tumor progression and survival in variety of cancers (Guo et al., 2015; Han et al., 2021; Kang et al., 2024; Zhang et al., 2021). CD34, as a vascular endothelial cell marker with high specificity and sensitivity, can directly reflect MV production. However, research on tumor lymphangiogenesis lags behind that of angiogenesis, but an increasing number of researchers believe that lymphangiogenesis also plays a more important role in tumor proliferation. The LV-specific marker D2-40 is more sensitive and specific than other LV endothelial markers for identifying LV endothelial cells (Heabah, Darwish & Eid, 2023). In the present study, we used CD34 and D2-40 to evaluate MVD and LVD in ESCC patients, finding that ESCC tissues had higher MVD and LVD than CAN tissues. This phenomenon is also present in colon cancer, breast cancer and esophageal adenocarcinoma (Lv et al., 2016; Martin, Rakha & Storr, 2022; Schiefer, Schoppmann & Birner, 2016), suggesting that angiogenesis and lymphangiogenesis are necessary for oxygen, nutrient, and metabolite transportation in tumors. To a large extent, the emergence of these new MVs and LVs enhances tumor growth activity. In essence, these studies have revealed a close association between that MVs, LVs, and the occurrence of solid tumors, serving as an inevitable indicator of tumor progression.

MVs and LVs play a pivotal role in tumorigenesis and tumor progression. Our study discovered a strong correlation between elevated MVD and LVD in patients with ESCC and the presence of lymphatic metastasis as well as advanced clinical staging. Other scholars have also highlighted the importance of LVD in the progression of epithelial-derived malignancies, with findings akin to our study, indicating that increased LVD is closely linked to tumor lymphatic metastasis (Chen et al., 2021; Kigure et al., 2013; Yu et al., 2011). Lymphatic metastasis represents the primary mode of ESCA metastasis and a key contributor to the unfavorable prognosis associated with ESCA (Inoue et al., 2008; Zeng et al., 2021). The formation of new blood vessels is crucial for cancer cell metastasis, facilitating direct tumor spread to target organs and serving as a vital conduit for tumor advancement. This process may serve as a crucial factor in ESCC metastasis among patients. Studies on the the combined synergistic effects of MVs and LVs in cancer progression is limited. The value of evaluating ESCC clinical progression based on the combined synergistic effect of MVD and LVD remains uncertain. Our study delves into the correlation between MVD and LVD in ESCC patients, revealing a positive linear relationship between the two. Furthermore, high MVD and LVD levels were found to be more significant in lymphatic metastasis and clinical staging compared to cases with low MVD and LVD. Elevated MVD and LVD levels together suggest potential associations with ESCC staging and lymphatic metastasis, serving as pivotal parameters for assessing the biological behavior and metastatic tendencies of ESCC in patients.

MVD and LVD play crucial role in tumor development and progression. Therefore, we investigated whether they could also serve as prognostic indicators. Our findings revealed that high MVD or high LVD were significantly linked to poor prognosis in patients with ESCC. It is worth noting that the combination of the two markers is more valuable for evaluating the prognosis of patients with ESCC. Notably, we observed that assessing both markers in combination yielded more meaningful prognostic information for ESCC patients, underscoring the significance of dual-marker evaluation. Furthermore, our univariate analysis indicated a strong association between high MVD and high LVD and unfavorable prognosis. Consistent with our study, high MVD in esophageal cancer has been significantly associated with poor prognosis (Schiefer, Schoppmann & Birner, 2016), and high LVD has been associated with poor prognosis in breast cancer (Zhang et al., 2017). Conversely, low MVD in renal cell carcinoma and pancreatic endocrine tumors may indicate a poorer prognosis (Marion-Audibert et al., 2003; Yildiz et al., 2008). This variance could be attributed to reduced MVD in renal cell carcinoma, indicating inadequate vascularization to support tumor growth, leading to necrosis, morphological changes, and worsened prognosis. The relationship between MVD and prognosis varies across different malignances, influenced by tumor evolution.

A limitation of our study is the inherent subjectivity in assessing microvessel and lymphatic vessel counts. To mitigate this bias, we engaged two expert pathologists to conduct evaluations and averaged counts from five hot spot areas, minimizing subjectivity-related errors.

As our understanding of the relationship between microvascular/lymphatic vessels, and tumor invasion/metastasis deepens, treatments targeting angiogenesis and lymphangiogenesis offer novel strategies in the fight against cancer. Emerging targets in these therapies have shown promise in inhibiting tumor growth, heralding a promising future for their application in oncology.

Conclusions

High MVD and LVD correlate with increased invasiveness, indicating a poorer prognosis for patients with ESCC. Evaluating MVD and LVD in combination holds greater prognostic value for ESCC patients, highlighting the importance of targeting tumor microvessels and lymphangiogenesis in ESCC treatment.

Supplemental Information

Supplemental Information 1 Data and code.

Additional Information and Declarations

Competing Interests

Author Contributions

Human Ethics

Data Availability

The authors declare that they have no competing interests.

Li Wen Qi conceived and designed the experiments, performed the experiments, prepared figures and/or tables, authored or reviewed drafts of the article, and approved the final draft.

Yu Fang Xie conceived and designed the experiments, performed the experiments, prepared figures and/or tables, authored or reviewed drafts of the article, and approved the final draft.

Wei Nan Wang conceived and designed the experiments, performed the experiments, prepared figures and/or tables, authored or reviewed drafts of the article, and approved the final draft.

Jia Liu conceived and designed the experiments, performed the experiments, analyzed the data, prepared figures and/or tables, authored or reviewed drafts of the article, and approved the final draft.

Kai Ge Yang analyzed the data, authored or reviewed drafts of the article, and approved the final draft.

Kai Chen analyzed the data, authored or reviewed drafts of the article, and approved the final draft.

Cheng Hua Luo analyzed the data, authored or reviewed drafts of the article, and approved the final draft.

Jing Fei analyzed the data, authored or reviewed drafts of the article, and approved the final draft.

Jian Ming Hu conceived and designed the experiments, analyzed the data, prepared figures and/or tables, authored or reviewed drafts of the article, and approved the final draft.

The following information was supplied relating to ethical approvals (i.e., approving body and any reference numbers):

This study was approved by the Ethics Committee of The First Affiliated Hospital, Shihezi University School of Medicine (Ethical Application Ref: 201803801), and informed consent was obtained from all participants.

The following information was supplied regarding data availability:

The raw data are available in the Supplemental File.

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
