# Peer review of "High microvessel and lymphatic vessel density predict poor prognosis in patients with esophageal squamous cell carcinoma"

_PeerJ, doi:10.7717/peerj.18080_

## Round 0.1 · original submission · Minor Revisions

1. The manuscript is relevant and exciting. Nevertheless, some issues remain that need to be solved.
2. The manuscript needs revision from an English native or an English language Editing Service. When you resubmit the manuscript, please include the action made to solve this issue.
3. Please include the sample origin (human or tissue samples) in the abstract's methods section.
4. Please include the microscope brand and the laser used in the method section.
5. In Figures 1 and 2, please indicate the MVD or LVD location in the tissue using arrows.
6. In tables 1-3, please include all acronyms I. e., TNM, HR.
7. Can you explain how a Chinese university can access Kasakh tissue samples?
8. The author must include the study's limitations in the discussion section.
9. The authors should resubmit the manuscript, including a point-by-pinot response to each reviewer.

Reviewer 1 ·

Basic reporting

There are many typos and certain parts of the writing that need improvement before publication. For instance, lines 69-74 and lines 213-214 require attention.

Experimental design

Additionally, the statement in lines 87-88 is incorrect. Patients who stopped follow-up should be labeled as censored (0) and should not be mixed with deceased patients (1) in the Kaplan-Meier plot analysis.

The authors mention Kazakh patients in the results section and Figure 5; however, the source of this patient cohort is not specified in the Materials and Methods section. Is it from a public online database?

Validity of the findings

Furthermore, in Figures 1 and 2, all photos should include a scale bar to indicate the length in µm.

Additional comments

NA

Reviewer 2 ·

Basic reporting

Improvement of Language and Clarity:
Although the manuscript is well-written, minor grammatical errors and awkward phrasings should be corrected. Engaging a professional editor for proofreading could enhance readability and overall presentation.

Ensure that all figures and tables are clearly labeled and easily interpretable. Some figures might benefit from additional annotations or more descriptive legends.

The manuscript provides a comprehensive background and sufficient literature references to support the context and significance of the study. The citations are relevant and up-to-date.

The article is well-structured, with appropriate sections and logical flow. The figures and tables are relevant and effectively illustrate the key findings.

The manuscript is self-contained and presents relevant results that address the hypotheses. The conclusions drawn are supported by the data presented.

Experimental design

The research is original and falls within the aims and scope of the journal. It addresses a significant gap in the understanding of microangiogenesis and lymphangiogenesis in esophageal squamous cell carcinoma (ESCC).

Inclusion and Exclusion Criteria:
Provide more detailed information on the inclusion and exclusion criteria for patient selection. This will enhance the reproducibility of the study and allow for a better understanding of the patient population.

Standardization of Immunoreactivity Scoring:
Include details on how immunoreactivity scoring was standardized and validated. This could involve descriptions of inter-observer reliability checks or calibration procedures.

Ethical Considerations:
Although the study mentions ethical approval and informed consent, a more detailed description of how ethical standards were maintained throughout the study would be beneficial.

The research question is clearly defined, relevant, and meaningful. The manuscript effectively explains how the study fills a gap in the existing knowledge regarding the prognostic significance of microvessel and lymphatic vessel density in ESCC.

The investigation is rigorous and meets high technical and ethical standards. The study includes appropriate controls and uses validated methods.

Validity of the findings

The manuscript demonstrates the impact and novelty of the findings by showing the prognostic value of microvessel and lymphatic vessel density in ESCC. The rationale and benefit of replication are clearly stated, emphasizing the study's contribution to the literature.

Mechanistic Insights:
Provide more detailed mechanistic insights into how increased MVD and LVD contribute to poorer prognosis. This could involve a discussion of potential molecular pathways or interactions with other prognostic factors.

All underlying data are provided and appear robust, statistically sound, and well-controlled. The statistical analyses are appropriate and support the conclusions drawn.

The conclusions are well-stated and directly linked to the original research question. They are appropriately limited to the supporting results, reinforcing the study's findings.

Additional comments

Discussion of Limitations:
The limitations section could be expanded to discuss potential biases or confounding variables in more detail. Addressing how these limitations might affect the study’s conclusions and suggesting ways to mitigate them in future research would be beneficial.

Future Directions:
Suggest specific future research directions based on the study’s findings. For instance, proposing targeted therapies that inhibit angiogenesis and lymphangiogenesis in ESCC or exploring the role of combined MVD and LVD as biomarkers in other cancers could be valuable.

---

## Round 0.2 · accepted · Accept

Dear Authors

The manuscript, entitled ¨High microvessel and lymphatic vessel density predict poor prognosis in patients with esophageal squamous cell carcinoma¨ is ready for publication.

Thanks for addressing all the reviewer`s concerns.
Best regards

Reviewer 1 ·

Basic reporting

NA

Experimental design

NA

Validity of the findings

NA

Additional comments

The authors have addressed most of my concerns.